# Association of *TP53* Single Nucleotide Polymorphisms with Prostate Cancer in a Racially Diverse Cohort of Men

**DOI:** 10.3390/biomedicines11051404

**Published:** 2023-05-09

**Authors:** Allison Duncan, Darryl Nousome, Randy Ricks, Huai-Ching Kuo, Lakshmi Ravindranath, Albert Dobi, Jennifer Cullen, Shiv Srivastava, Gregory T. Chesnut, Gyorgy Petrovics, Indu Kohaar

**Affiliations:** 1Center for Prostate Disease Research, Murtha Cancer Center Research Program, Department of Surgery, Uniformed Services University of the Health Sciences, Bethesda, MD 20817, USA; akduncan923@gmail.com (A.D.); dnousome@gmail.com (D.N.); randyokang@gmail.com (R.R.); ckuo@idcrp.org (H.-C.K.); lravindranath@cpdr.org (L.R.); adobi@cpdr.org (A.D.); jxc1650@case.edu (J.C.); shsr629@gmail.com (S.S.); gregory.t.chesnut.mil@health.mil (G.T.C.); gpetrovics@cpdr.org (G.P.); 2F. Edward Hebert School of Medicine, Uniformed Services University of the Health Sciences, Bethesda, MD 20814, USA; 3Henry M. Jackson Foundation for the Advancement of Military Medicine Inc., Bethesda, MD 20817, USA; 4Urology Service, Walter Reed National Military Medical Center, Bethesda, MD 20814, USA

**Keywords:** prostate cancer, single nucleotide polymorphism, SNP, *TP53*, Pro47Ser, Arg72Pro, African American, medical disparity

## Abstract

Growing evidence indicates the involvement of a genetic component in prostate cancer (CaP) susceptibility and clinical severity. Studies have reported the role of germline mutations and single nucleotide polymorphisms (SNPs) of *TP53* as possible risk factors for cancer development. In this single institutional retrospective study, we identified common SNPs in the *TP53* gene in AA and CA men and performed association analyses for functional *TP53* SNPs with the clinico-pathological features of CaP. The SNP genotyping analysis of the final cohort of 308 men (212 AA; 95 CA) identified 74 SNPs in the *TP53* region, with a minor allele frequency (MAF) of at least 1%. Two SNPs were non-synonymous in the exonic region of *TP53*: rs1800371 (Pro47Ser) and rs1042522 (Arg72Pro). The Pro47Ser variant had an MAF of 0.01 in AA but was not detected in CA. Arg72Pro was the most common SNP, with an MAF of 0.50 (0.41 in AA; 0.68 in CA). Arg72Pro was associated with a shorter time to biochemical recurrence (BCR) (*p* = 0.046; HR = 1.52). The study demonstrated ancestral differences in the allele frequencies of the *TP53* Arg72Pro and Pro47Ser SNPs, providing a valuable framework for evaluating CaP disparities among AA and CA men.

## 1. Introduction

Prostate cancer (CaP) is the most common non-skin cancer and the second leading cause of cancer death in American men [1]. It has an unequal burden of disease incidence and mortality among men of different ancestries [2]. Prostate cancer is a clinically heterogeneous disease consisting of diverse clinico-pathologic and progression features, and it is characterized by a large subset of the indolent cancer type. Therefore, it is critical to identify molecular and genetic markers with diagnostic, predictive, and prognostic potential in addition to the standard of care (SOC) variables [3]. A significant proportion of CaP susceptibility has been attributed to inherited predisposition. Genome-wide association studies (GWAS) have identified 269 low-penetrance single nucleotide polymorphisms (SNPs) associated with CaP risk or aggressiveness [4]. The *TP53* gene, also referred to as the guardian of the genome, is a key tumor suppressor gene involved in DNA repair, cell cycle arrest, apoptosis, cell metabolism, and genomic stability [5,6]. *TP53* is the most commonly mutated gene in human cancers, and malfunction of the *TP53* pathway is a key feature of human tumors [7]. Germline mutations in *TP53* predispose people to Li–Fraumeni syndrome (LFS), which is characterized by early onset of cancers, including breast cancer, sarcomas, brain tumors, leukemia, and adrenocortical carcinomas [8,9,10]. However, the role of germline mutations in *TP53* is not completely understood in CaP. Some studies have pointed to the role of specific *TP53* SNPs as a possible risk factor for CaP [11,12]. Polymorphisms in codons 47 and 72 in the *TP53* tumor suppressor gene have been shown to affect phosphorylation and the expression of pro-apoptotic genes, respectively, both of which could increase cancer risk [13]. A more recent study showed that heterozygosity at codon 72 SNP (rs1042522, GC) and either of the two intronic SNPs rs9895829(TC) and rs2909430(AG) are associated with up to a five-fold greater CaP risk, greater tumor-promoting inflammation, and shorter patient survival [14].

Racial disparities in CaP incidence and mortality rates remain significant, even after adjusting for socioeconomic status and access to healthcare, which is suggestive of a disproportionate contribution from genetic factors [15]. Several Veterans Affairs studies have revealed conflicting results in CaP outcomes for black and white men within a single, equal-access healthcare system [16,17]. Black men have an increased risk of CaP, especially clinically aggressive CaP, and elevated PSA levels at the time of diagnosis compared to white men in similar cohorts. Genetic associations between such clinical findings and SNPs have been described earlier, but they have shown inconsistent results. Thus, further understanding and additional confirmation of such genetic factors may be beneficial in understanding clinical outcomes, which could influence screening and treatment guidelines across AA and CA men in an equal-access military healthcare system.

We hypothesized that germline variants in *TP53* could contribute to the distinct clinical phenotypes of CaP. Thus, the goals of this study were to identify the common SNPs in the *TP53* gene in AA and CA men and to perform an association analysis for functional variants in *TP53* with the clinico-pathological features of CaP, including biochemical recurrence-free survival and metastasis-free survival.

## 2. Materials and Methods

### 2.1. Study Population

This study was based on a retrospective cohort design utilizing germline blood DNA derived from OncoArray data from 321 men (AA = 216 and CA = 105) with biopsy-confirmed CaP who underwent RP treatment at Walter Reed National Military Medical Center (WRNMMC). Patients were only included if they consented to enrollment in the CPDR biospecimen databank and multicenter national clinical database. The databases have been approved by the Institutional Review Boards (IRBs) at the WRNMMC and the Uniformed Services University of the Health Sciences (USUHS) in Bethesda, Maryland. All subjects provided written informed consent and agreed to participate in the study under IRB-approved protocols. These patients were under the equal-access DoD healthcare system, underwent radical prostatectomy for localized CaP within the first year of diagnosis, and were clinically followed up to 25 years. Demographics, clinical characteristics, pathology information, and all treatment data were obtained from the CPDR multicenter national clinical database. Patient characteristics included patient age at CaP diagnosis (years), self-reported race (AA or CA), diagnostic PSA (ng/mL), pathologic T Stage (pT2, T3, T4), biopsy and pathologic Gleason scores (≤6, 3 + 4, 4 + 3, and 8–10), biochemical recurrence (BCR), time to BCR, and metastasis outcome. A BCR event was defined as either two successive post-RP PSAs of ≥0.2 ng/mL or initiation of salvage therapy after a rising PSA of ≥0.1 ng/mL. Patients who underwent salvage therapy without a rising PSA level of ≥0.1 were categorized as having had a non-BCR event and were censored at the date of the initiation of salvage therapy. Patients who showed an initial PSA level of ≥0.2 ng/mL but no confirmatory PSA ≥ 0.2 ng/mL and no initiation of salvage therapy were categorized as having had a non-BCR event and were censored at the last known date of PSA < 0.2 ng/mL. Patients who were lost to follow-up or who died without any evidence of BCR were censored as non-events at the date of their last known clinic visit or date of death, respectively [18]. The study exclusion criteria included men without self-reported race, distant metastasis within a year, local and/or distant metastases at biopsy, distant metastasis at pathology, and men who underwent neoadjuvant therapy. The clinical characteristics of the patients are shown in Table 1.

### 2.2. SNP Analysis and Validation

Genotype analysis for common SNPs with a minor allele frequency (MAF) of at least 0.01 was performed using Eagle [19] for phasing, and for imputation, TOPMed (v R2 on GRCh38) was used as a reference comprising 194,512 haplotypes [20]. Genotype data were split into 20 MB regions with an overlap of 5 MB. Experimental validation for the imputed SNPs (rs1800371/Pro47Ser and rs1042522/Arg72Pro) was performed using the droplet digital polymerase chain (ddPCR) approach with a QX200 (BioRad). Briefly, a ddPCR mastermix was prepared containing 11 μL 2X ddPCR Supermix (BioRad), 1.1 μL 20X TaqMan SNP Genotyping Assay (BioRad, ThermoFisher Scientific, CA, USA; Appendix A), and 7.9 μL nuclease-free water (Qiagen) per sample. The mastermix was prepared at room temperature, and 20 μL was added to 2 μL (=5 ng) of each DNA sample. Samples were loaded into individual wells of DG8TM cartridges (BioRad), and droplets were generated using a QX200 Droplet Generator (BioRad). For each sample, 40 μL of droplet mix was then transferred to a 96-well plate, and PCR was performed in a thermal cycler using the following cycling conditions: 95 °C × 10 min; 40 cycles of (94 °C × 30 s, 60 °C × 60 s); 98 °C × 10 s; and 40 °C × 10 min. A BioRad QX200 Droplet Reader was then used to assess the droplets as positive or negative based on fluorescence amplitude. QuantaSoft software (v1.7.4.0917) (BioRad) was used to analyze the droplet data.

### 2.3. Statistical Analysis

Descriptive distributions for clinico-pathological variables in the overall cohort stratified by exonic SNP genotypes were examined. The chi-square test or Fisher’s exact test were used to compare categorical variables, and the Mann–Whitney U test was used to compare continuous variables. Time to biochemical recurrence (BCR) or metastasis was evaluated using Cox proportional hazards (PH) modeling, adjusting for age at diagnosis, race, and Gleason score, to examine SNP effects. Unadjusted Kaplan–Meier survival analysis and log-rank testing were used to show the probability of BCR-free survival stratified by genotypes. All statistical analysis was performed using SAS version 9.4 (SAS Institute Inc., Cary, NC, USA), and statistical significance was set at *p* < 0.05.

## 3. Results

The clinical and pathological features of the patient cohort are summarized in Table 1. The cohort had a higher proportion of AA patients compared to CA patients (69% vs. 31%) and was over-represented by subjects with serum PSA in the range of 4–9 ng/mL (63%), T2 clinical stage (74%), and pathological Gleason score ≤ 7 (83%). Post-prostatectomy follow-up data showed that 15% of the subjects had biochemical recurrence, with 3% having metastasis. 

SNP analysis of the final cohort of 308 patients with available clinical data identified 74 SNPs in the *TP53* gene region that had an MAF of at least 1%. A description of the 74 SNPs is included in Appendix A. Two imputed SNPs (rs1800371/Pro47Ser and rs1042522/Arg72Pro) of the 74 SNPs were non-synonymous, functional SNPs located in exon 4 of *TP53* (Figure 1). The MAFs for Pro47Ser and Arg72Pro were 0.01 and 0.50, respectively. Both these imputed SNPs were experimentally validated with the ddPCR approach (Figure 2). The concordance between the TaqMan genotypes and imputed genotypes was 99%. 

Race-stratified analysis of the SNPs demonstrated and confirmed ancestry-dependent differences in the allele frequencies of *TP53* Arg72Pro and *TP53* Pro47Ser SNPs in AA and CA CaP (Table 2; Appendix A). The Pro47Ser variant had an MAF of 0.01 in AA men but was not detected in CA men, while Arg72Pro was the most common SNP, with an MAF of 0.50 (AA, 0.41; CA, 0.68) (Appendix A). The MAF distribution in our dataset was consistent with publicly available databases: Pro47Ser variant A = 0.005591/28 (1000 Genomes) and A = 0.02439/8 (HapMap); Arg72Pro G = 0.457069/2289 (1000 Genomes) and G = 0.43865/143 (HapMap). 

An association analysis of the *TP53* Pro47Ser and Arg72Pro SNPs with respect to the clinico-pathologic features of CaP is shown in Table 2. Except for race, there were no observed statistically significant associations between the Arg72Pro or Pro47Ser SNPs and age of onset of prostate cancer, CaP aggressiveness (grade, clinical stage), tumor upgrading from biopsy to RP (Appendix A), or disease outcome (BCR, Metastasis). However, Arg72Pro SNP was the only exonic SNP significantly (*p* = 0.046; HR = 1.52) associated with a shorter time to BCR following radical prostatectomy, after adjusting for patient age at diagnosis, race, and Gleason score (Table 3). Additionally, we found that three non-exonic SNPs were significantly associated with a shorter time to BCR (rs4968186, intergenic, *p* 0.004, HR 1.75; rs9894227, intronic, *p* 0.007, HR 1.72; and rs8079544, intronic, *p* 0.02, HR 1.63). 

## 4. Discussion

The present study investigated the association between common functional germline variants in the *TP53* gene and CaP in a racially diverse cohort of men using longitudinal clinical follow-up. We confirmed ancestry-dependent differences for MAFs across AA and CA men and found that Arg72Pro was associated with a shorter time to BCR. Arg72Pro and Pro47Ser SNPs have been well characterized for their function and association with cancer [12,21,22,23,24,25]; however, there is no consistent conclusion for CaP risk or clinico-pathological features [26,27,28,29,30]. This variability may be attributed to small sample size, study design, and mixed ancestry. 

Additionally, a prognostic role for SNPs in relation to disease and treatment outcomes has been found in many cancers, including CaP [31], lung cancer [32], gastric cancer [33], breast cancer [34], and acute lymphoblastic leukemia. One common variant, Arg72Pro, leads to altered activities by the TP53 protein. The C to G base change results in a proline (Pro) to arginine (Arg) amino acid alteration in the proline-rich domain, which impacts programmed cell death or cell cycle arrest. Additionally, TP53 arginine has an increased affinity for MDM-2, resulting in destabilization of the protein, lower levels of the TP53 protein, and, ultimately, earlier onset of TP53-associated tumors [14]. On the other hand, a less frequent and AA-centric Pro47Ser variant exhibits functional consequences by attenuating p53 transcriptional activation and pro-apoptotic functions [35]. 

The inactivation of p53 via somatic and germline *TP53* mutations has been documented for both aggressive disease and poor outcomes in many cancer types [36]. A meta-analysis of breast cancer case–control studies (7841 cases and 8876 controls) found a significant association between the Arg72Pro SNP and the risk of breast cancer. Race/ethnicity-stratified analysis found that the Pro allele was associated with the risk of breast cancer in Caucasians for the dominant model and additive model (*p* = 0.02) and in Africans for the recessive model and additive model (*p* = 0.03) [37]. Similar findings were reported by two other meta-analyses. The first included 11 studies with 950 cases and 882 controls in the Asian population [38], while the second included the Indian population and covered seven studies with 1249 cases and 1838 controls [39]. On the contrary, a meta-analysis study based on 18,718 cases and 21,261 controls showed that the *TP53* Arg72Pro polymorphism is associated with an increased risk of cancer in Asians and Caucasian Americans only and is not associated with cancer risk in other populations [25]. Another study on Caucasian patients based on 90 breast and 162 colorectal cancer patients revealed that the Arg72Pro SNP may affect the function of *TP53* mutations in breast carcinomas but not in colorectal carcinomas [40]. Although the presence of both Arg and Pro has been revealed in the context of different cancers, multiple studies have conflicting results on the relationship between allele selection, SNPs, and disease risk. This indicates that SNP prevalence is largely influenced by the racial composition of the study cohorts [31,32,33,34,35,36,37,38,39,40,41,42,43,44]. Additional factors contributing to the variability may be explained by the types of allelic variants, cohort size, and eligibility criteria for studies included under meta-analyses.

Somatic mutations of *TP53* have also been reported with variable frequencies in primary CaP and consistently with higher frequencies in metastatic disease [45]. Several studies from our center over the years, including a recent study, have shown that focal alterations of the TP53 protein in prostatectomy specimens are associated with BCR and metastatic progression [46].

A recent multi-institutional retrospective study conducted by Maxwell et al. on CaP incidence in a cohort of patients with Li–Fraumeni syndrome (LFS) showed that men with LFS had a 25-fold increased risk of CaP compared to the population controls, and the rate of inherited deleterious *TP53* variants was nine times higher than men with no CaP [47]. Our current findings on the association between *TP53* SNPs and aggressive disease further support the role of germline *TP53* functional alterations in prostate tumorigenesis and suggest a potential clinical benefit of including an evaluation of germline *TP53* variants (SNPs identified in this study and pathogenic SNVs) in CaP susceptibility testing, especially in the context of high-risk men [48,49].

In addition to the genetic influence on CaP, studies have described differences in such associations as they pertain to patient ancestry [50]. While associations between AA race, *TP53* SNPs, and CaP incidence, mortality, and post-treatment BCR have been made in the past, the three primary guideline tools—National Comprehensive Cancer Network (NCCN), United States Preventive Services Task Force (USPSTF), and American Urological Association (AUA) Clinical Guidelines—used by physicians to guide patient screening and treatment do not suggest their use in treatment decision-making recommendations, despite their clinical relevance [51,52,53]. 

The findings in our study not only further confirm previous research but also improve the understanding of the association between *TP53* SNPs, race/ancestry, and poor disease outcomes. The present study provides important data that could influence medical providers in improving screening tools, germline CaP susceptibility testing for high-risk men, and treatment approaches for patients. The limitations of this study include self-reported race [54] and the limited generalizability of our study cohort compared to other US cohorts. Future directions include a comprehensive analysis of germline variants in the *TP53* pathway in high-risk men, an investigation of the *TP53* mutation profile to predict response to treatment, and a functional understanding of the biology of these SNPs in relation to aggressive CaP in a patient cohort in which ancestry is genetically confirmed. Future efforts should also consider the potential contribution of genetics, ethnicity/race, and socioeconomic factors to the CaP health disparity.

## 5. Conclusions

A comprehensive analysis of common germline variants in *TP53* confirmed genetic heterogeneity across AA and CA men for Arg72Pro and Pro47Ser variants and suggested that *TP53* Arg72Pro predisposes men to clinically significant prostate cancer. Therefore, *TP53* should be included in the CaP germline screening panel for high-risk men representing different ancestries. 

## Figures and Tables

**Figure 1 biomedicines-11-01404-f001:**
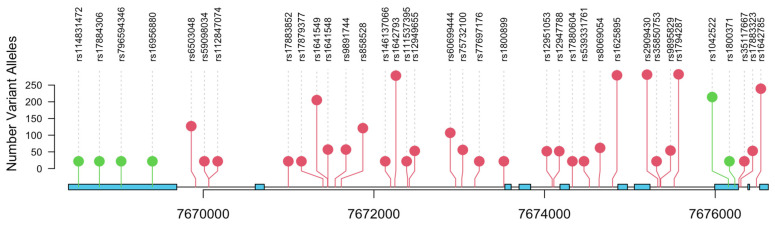
Lollipop plot for single nucleotide polymorphisms in *TP53*. Variants in the coding region of the *TP53* gene are indicated in green. Exons and UTRs are shown in blue.

**Figure 2 biomedicines-11-01404-f002:**
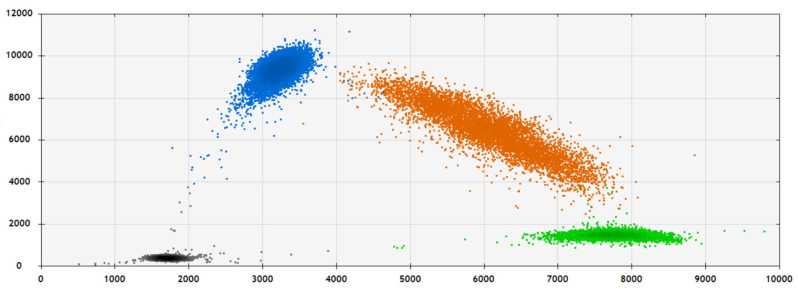
Representative graph of SNP genotyping for rs1800371 (C/G) using the digital droplet PCR (ddPCR) approach. 2-D amplitude view in which each axis represents the amplitude of fluorescence for either FAM (vertical axis) or VIC (horizontal axis). The FAM probe can hybridize only to the alternate allele (G allele), while the VIC probe hybridizes only to the reference allele (C allele).

**Table 1 biomedicines-11-01404-t001:** Descriptive statistics of the patient cohort (N = 308).

Characteristic	N = 308 (%)
Diagnosis Age (in years)	57 (9)
Unknown	1
Race	
African American	212 (69%)
Caucasian	95 (31%)
Unknown	1
Diagnosis PSA (ng/mL)	
1: <4	72 (24%)
2: 4–9	193 (63%)
3: 10–20	31 (10%)
4: >20	10 (3.3%)
Unknown	2
Pathologic T Stage	
T2	228 (74%)
T3–4	80 (26%)
Biopsy Gleason score	
≤6	204 (70%)
7	69 (24%)
8–10	17 (5.9%)
Unknown	18
Pathologic Gleason score	
3 + 3	169 (57%)
3 + 4	76 (26%)
4 + 3	24 (8.2%)
8–10	25 (8.5%)
Unknown	14
BCR	47 (15%)
Unknown	3
Metastasis	9 (2.9%)

**Table 2 biomedicines-11-01404-t002:** TP53 rs1800371 and rs1042522 SNP association analysis.

SNP	TP53 rs1800371 SNP			TP53 rs1042522 SNP			
Patient Characteristics	GG, N = 302	GA, N = 6	*p*-Value	GG, N = 87	GC, N = 137	CC, N = 84	*p*-value
Diagnosis Age	57 (9)	56 (12)	0.6	58 (9)	57 (8)	57 (9)	0.8
Unknown	1	0		0	1	0	
Race			0.2				<0.001
African American	206 (68%)	6 (100%)		81 (93%)	87 (64%)	44 (52%)	
Caucasian	95 (32%)	0 (0%)		6 (6.9%)	49 (36%)	40 (48%)	
Unknown	1	0		0	1	0	
Diagnosis PSA			0.085				0.8
1: <4	70 (23%)	2 (33%)		22 (25%)	28 (21%)	22 (26%)	
2: 4–9	191 (64%)	2 (33%)		55 (63%)	87 (64%)	51 (61%)	
3: 10–20	30 (10%)	1 (17%)		9 (10%)	14 (10%)	8 (9.5%)	
4: >20	9 (3.0%)	1 (17%)		1 (1.1%)	6 (4.4%)	3 (3.6%)	
Unknown	2	0		0	2	0	
Pathologic T Stage			>0.9				0.6
T2	223 (74%)	5 (83%)		57 (68%)	104 (76%)	59 (70%)	
T3–4	79 (26%)	1 (17%)		22 (25%)	33 (24%)	25 (30%)	
Biopsy Gleason score			0.2				0.8
≤6	199 (70%)	5 (83%)		57 (68%)	93 (73%)	54 (69%)	
7	69 (24%)	0 (0%)		21 (25%)	27 (21%)	21 (27%)	
8–10	16 (5.6%)	1 (17%)		6 (7.1%)	8 (6.2%)	3 (3.8%)	
Unknown	18	0		3	9	6	
Pathologic Gleason score			0.6				>0.9
3 + 3	164 (57%)	5 (100%)		46 (55%)	74 (57%)	49 (60%)	
3 + 4	76 (26%)	0 (0%)		22 (26%)	35 (27%)	19 (23%)	
4 + 3	24 (8.3%)	0 (0%)		7 (8.3%)	10 (7.8%)	7 (8.6%)	
8–10	25 (8.7%)	0 (0%)		9 (11%)	10 (7.8%)	6 (7.4%)	
Unknown	13	1		3	8	3	
BCR	47 (16%)	0 (0%)	0.6	13 (15%)	20 (15%)	14 (17%)	>0.9
Unknown	3	0		1	1	1	
Metastasis	8 (2.6%)	1 (17%)	0.2	3 (3.4%)	6 (4.4%)	0 (0%)	0.2

**Table 3 biomedicines-11-01404-t003:** Association of *TP53* SNPs with time to biochemical recurrence.

Position	SNP	Location	HR	*p*-Value
chr17:7667612:A:G	rs4968186	downstream	1.753	0.004
chr17:7676963:G:A	rs9894227	intronic	1.720	0.007
chr17:7676734:C:T	rs8079544	intronic	1.632	0.020
chr17:7676154:G:C	rs1042522	exonic	1.523	0.046
chr17:7664197:C:T	rs35119871	intergenic	1.461	0.046
chr17:7675519:A:G	rs1794287	intronic	1.508	0.058
chr17:7676230:G:A	rs1800371	exonic	1.473	0.067

Time to event (BCR) was included in a Cox PH model adjusting for age at diagnosis, race, and Gleason score to examine SNP effects.

## Data Availability

Data supporting reported results can be found in Table 1, Table 2 and Table 3 and Appendix A.

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
