# Peer review of "Association of TP53 Single Nucleotide Polymorphisms with Prostate Cancer in a Racially Diverse Cohort of Men"

_biomedicines, 2023, doi:10.3390/biomedicines11051404_

Round 1

Reviewer 1 Report

The manuscript “Association of TP53 Single Nucleotide Polymorphisms with Prostate Cancer in a Racially Diverse Cohort of Men” by Allison Duncan et al focuses on an important issue of a race-based health disparities and their under-looked genetic aspects in prostate cancer (CaP). Addressing such aspects is an integral part of the personalized medicine approach. In particular, the authors studied the association of common SNPs in the TP53 gene with clinico-pathological features of CaP in African American and Caucasian individuals. The authors found an association of Arg72Pro SNP of Tp53 with shorter time to biochemical recurrence, which may have a clinical value.

The study methodology is sound, the study cohort is big enough to draw a preliminary conclusion, however it should be expanded in future.

Notably, the ancestry/race of the study participants was self-reported, and there are some limitations of such approach (https://humgenomics.biomedcentral.com/articles/10.1186/s40246-014-0023-x). I recommend to mention these limitations in the text of the manuscript.

The authors may also want to discuss in details role of Arg72Pro SNP in other cancers (colorectal, breast, etc) with particular focus on studies involving study participants of AA and CA ancestries (https://bmcmedgenet.biomedcentral.com/articles/10.1186/s12881-020-01133-8 and others).

Author Response

Reviewer 1:

The manuscript “Association of TP53 Single Nucleotide Polymorphisms with Prostate Cancer in a Racially Diverse Cohort of Men” by Allison Duncan et al focuses on an important issue of a race-based health disparities and their under-looked genetic aspects in prostate cancer (CaP). Addressing such aspects is an integral part of the personalized medicine approach. In particular, the authors studied the association of common SNPs in the TP53 gene with clinico-pathological features of CaP in African American and Caucasian individuals. The authors found an association of Arg72Pro SNP of Tp53 with shorter time to biochemical recurrence, which may have a clinical value.

The study methodology is sound, the study cohort is big enough to draw a preliminary conclusion, however it should be expanded in future.

Notably, the ancestry/race of the study participants was self-reported, and there are some limitations of such approach (https://humgenomics.biomedcentral.com/articles/10.1186/s40246-014-0023-x). I recommend to mention these limitations in the text of the manuscript.

We thank the reviewer for the kind feedback and important suggestions.

We are presently analyzing this cohort for somatic mutation signatures including TP53 using whole exome sequencing (WES) to explore the associations of germline variants with the somatic tumor genome on CaP outcomes in this racially diverse cohort of men. In relation to validation and establishing the clinical utility of these mutations for advanced CaP, we also plan to collaborate with researchers who have access to specimens from advanced/metastatic CaP, especially high-risk AA men.

Based on the reviewer’s suggestions, we have incorporated information on the limitation of self- reported ancestry and added relevant references in the manuscript (pages # 8). Thank you for raising this important point. However, we want to mention that a subset of subjects included in the study were analyzed for genetic ancestry using PCA analysis (Peddy program using 1000G as reference dataset), where concordance between self-reported and genetic ancestry was 96% (Kohaar et al, Nat Comm 2022). Please see the revised manuscript (in track changes) attached for reference.

The authors may also want to discuss in details role of Arg72Pro SNP in other cancers (colorectal, breast, etc) with particular focus on studies involving study participants of AA and CA ancestries (https://bmcmedgenet.biomedcentral.com/articles/10.1186/s12881-020-01133-8 and others).

As suggested by the reviewer, we have included the information on the role of Arg72Pro SNP in other cancers with emphasis on studies involving study participants of AA and CA ancestries in the manuscript (page # 7). Please see the revised manuscript (in track changes) is attached for reference.

Reviewer 2 Report

Manuscript entitled "Association of TP53 Single Nucleotide Polymorphisms with Prostate Cancer in a Racially Diverse Cohort of Men"

1. Since this work performed analysis on patient outcome, it is ,andatory to clear described the treatment protocol pf the patients and the treatment (TURP and or OP, with or without chemotherapy or targeted therapy, etc.) should also be listed into survival analysis.

2. The genotype should be validate by direct sequencing at least for some cases.

Acceptable

Author Response

Reviewer 2:

1.  Manuscript entitled "Association of TP53 Single Nucleotide Polymorphisms with Prostate Cancer in a Racially Diverse Cohort of Men"

Since this work performed analysis on patient outcome, it is ,mandatory to clear described the treatment protocol of the patients and the treatment (TURP and or OP, with or without chemotherapy or targeted therapy, etc.) should also be listed into survival analysis.

As suggested by the reviewer, we have updated the manuscript. This study is based on a retrospective cohort design utilizing germline blood DNA derived Oncoarray data from 321 men (AA = 216 and CA = 105) who were treated with radical prostatectomy for localized CaP at Walter Reed National Military Medical Center (WRNMMC). Blood was drawn prior to the surgery. Regarding prior treatment, the study included men with  biopsy‐confirmed CaP who underwent RP treatment. We have now clarified the study inclusion and exclusion including treatment profile in relation to outcome (BCR) in Page #2 and #3. Please see the revised manuscript attached in track changes.

2. The genotype should be validate by direct sequencing at least for some cases.

Experimental validation for the imputed SNPs (rs1800371/Pro47Ser and rs1042522/Arg72Pro) using Oncoarray data was performed using droplet digital polymerase chain (ddPCR) approach using a QX200 (BioRad). Concordance between TaqMan genotypes and imputed genotypes (from SNP array dataset) was 99%. The details have been highlighted on page# 4. Additionally, Kohaar et al, Nat Comm 2022 paper provides evidence of strong concordance (99.15%; 117 of 118) between ddPCR and WGS approaches. Please see the revised manuscript attached in track changes for reference.

Round 2

Reviewer 2 Report

The revision is acceptable in the current form.

Acceptable